# A Practical Trajectory Tracking Scheme for a Twin-Propeller Twin-Hull Unmanned Surface Vehicle

**Jiucai Jin** [1,2,*] **, Deqing Liu** [1,2] **, Dong Wang** [1,2] **and Yi Ma** [1,2]

1    First Institute of Oceanography, Ministry of Natural Resources, Qingdao 266061, China;
     liudeqing@fio.org.cn (D.L.); wangdong@fio.org.cn (D.W.); mayimail@fio.org.cn (Y.M.)
2    Ocean Telemetry Technology Innovation Center, Ministry of Natural Resources, Qingdao 266061, China
*    Correspondence: jinjiucai@fio.org.cn; Tel.: +86-159-5338-0245

**Abstract:** Trajectory tracking is a basis of motion control for Unmanned Surface Vehicles (USVs), which has been researched well for common USVs. The twin-propeller and twin-hull USV (TPTH-USV) is a special vehicle for applications due to its good stability and high load. We propose a three-layered architecture of trajectory tracking for the TPTH-USV which explicitly decomposes into trajectory guidance, a motion limitator and controller. The trajectory guidance transforms an expected trajectory into an expected speed and expected course in a kinematic layer. The motion limitator describes some restriction for motion features of the USV in the restriction layer, such as the maximum speed and maximum yaw rate. The controller is to control the speed and course of the USV in the kinetic layer. In the first layer, an adaptive line-of-sight guidance law is designed by regulating the speed and course to track a curved line considering the sideslip angle. In the second layer, the motion features are extracted from an identified speed and course coupled model. In the last layer, the course and speed controller are designed based on a twin-PID controller. The feasibility and practicability of the proposed trajectory tracking scheme is validated in sea experiments by a USV called 'Jiuhang 490'.

**Keywords:** trajectory tracking; unmanned surface vehicle; model identification; line-of-sight





## 1. Introduction

An Unmanned Surface Vehicle (USV) is a novel kind of multifunctional surface platform, which has been applied in many oceanic fields in recent years, such as ocean surveying, hydrology measurements, underwater acoustic communication, target tracking [1–3], etc. The motion control of USVs is a basic and essential part for autonomous operation, which has usually been inspired by conventional vehicles' control. In general, there are three issues in the motion control of vehicles, which contain point stabilization, path following and trajectory tracking. Point stabilization is used to stabilize the vehicle around an expected position, and path following is used to follow a predefined path for the vehicle, while trajectory tracking is used to track a predefined path with a time constraint. Path following and trajectory tracking have, recently, received considerable attention from the control communities, and many control methods have been applied, such as PID, fuzzy, backstepping, sliding mode control, evolutionary algorithms [4–6], etc.

The trajectory tracking of USVs can be departed into two categories, which are called direct and indirect control [7], and the first one is that the control issues are deemed as the zeroing of position errors, and the other is that the control issues are decomposed into guidance in the kinematic level and control in the kinetic level. In the direct control, the trajectory tracking is seen as a whole issue, and the stabilization control for tracking errors is designed based on a dynamic model of the USV, and lots of theories and methods have been developed [8–10]. Many control laws have been designed based on backstepping technology, and the stabilization is usually given out perfectly. However, the direct control



emerges mainly in theoretical research and is not convenient to be applied in the actual USVs due to their complexity [11].

### 1.1. Related Works

In the indirect control, the control issue is decomposed into guidance in the kinematic level and control in the kinetic level. In the kinematic level, the guidance law is designed by the speed and course control variables, while the speed and course control variables are deemed as expected values in the kinetic level. The kinematic control is equivalent to a work space control [12,13], where the work space (also known as the operational space) represents the physical space (environment) in which a vehicle moves. The kinematic level considers the geometrical aspects of motions purely, without reference to the forces and moments that generate such motions. The kinetic controllers consider how forces and moments generate the vehicle's motion, which are typically designed based on model-based methods.

Since the indirect control has an obvious physical meaning in path following and trajectory tracking, lots of works have been published and applied. The course and speed control for USVs is usually seen as the basic controller for indirect control, which has been researched broadly [14–16]. The line-of-sight guidance law is used broadly in a ship's trajectory tracking [17,18], and a time-varying look-ahead distance and integral LOS technology has been developed [19], which is used to solve the sideslip angle problem. Lots of LOS technologies have been applied in USVs' kinematic control [20,21]. A trajectory tracking controller for an underactuated USV with multiple uncertainties and input constraints has been designed based on indirect control, and the design process of the controller is simplified and easy to implement due to the guidance law in the kinematic level [7]. Defining a set of guidance laws at the kinematic level for an underactuated USV in a two-dimensional space, a nonlinear Lyapunov-based control law has been designed to yield the convergence of the path-following error coordinates to zero [11]. A modified LOS guidance algorithm has been proposed for the path following control of the underactuated USV, which can adaptively change the guidance law to respond to the longitudinal and lateral path following error [22]. Moreover, many algorithms have been derived by combining the traditional LOS technology and nonlinear control methods [23,24]. In addition, some novelty methods have been applied in the guidance law, such as bioinspired neural [25], deep reinforcement learning methods [26] and vector field [27]. The twin-propeller and twin-hull USV (TPTH-USV) is a usual vehicle for applications due to its good stability and high load [28], such as 'Springer' [29], 'JiuHang-490' [30].

Although many schemes of the trajectory tracking have been developed in the above works, most of the control laws cannot be directly or easily applied in universal USVs, and there are three reasons in view of practicability. Firstly, the control laws are too complicated to be used in actual engineering, also due to their high calculate costs. Secondly, the engineers could not understand the control laws well due to the complexity of the algorithms, and it is difficult to transfer the algorithms to executable procedures. Thirdly, most of the control laws are based on the dynamic models which are usually simplified for the actual systems, so the parameters and application condition of the controllers may not be suitable for common USVs. In summary, the control laws are usually designed for different vehicles and systems, and the bad-transplantation of the controllers appears in actual engineering due to their strong pertinence. In order to improve the disadvantages of the above trajectory tracking control, such as bad transplantation, compatibility for trajectory tracking and path following, a three-layered architecture of trajectory tracking for TPTH-USVs is proposed, and it is nearly suitable for the type of TPTH-USVs. The proposed scheme focuses on the design of guidance law for curved lines, and it is suitable for trajectory tracking and path following simultaneously by considering the speed variable.

*1.2. Scheme Design and Paper Structure*

Considering the above advantages and disadvantages of the indirect control, a three-layered architecture scheme for trajectory tracking for the TPTH-USV is designed which contains the kinematic layer, restriction layer and kinetic layer, which are shown in Figure 1:

1.  In the kinematic layer, an improved LOS law is proposed based on an adaptive look-ahead distance, which can not only steer the course of the USV, but can also regulate the speed of the USV.
2.  In the restriction layer, some constraint of control is given out based on an identified model. Since a precise model of the USV cannot be easily acquired due to the complicated hydrodynamic analysis and huge experimental cost, some constraints can be evaluated based on some classic model or basic experiment data.
3.  In the control level, a twin-PID controller is designed for the course and speed control, which is independent on the model and can be realized in the actual USV.

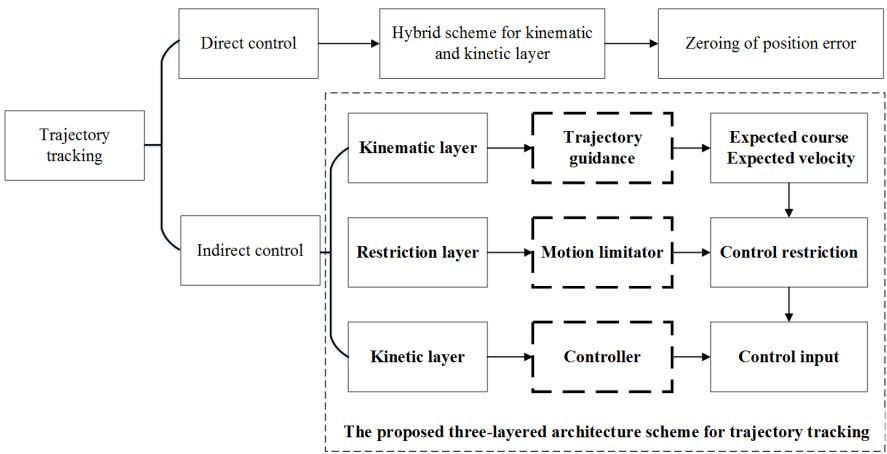

**Figure 1.** The two categories of the trajectory tracking and the proposed three-layered architecture scheme for trajectory tracking of the USV.

The advantages of the proposed algorithm can be illustrated as four aspects:

1.  The first one is that the improved LOS guidance law is suitable for all the USVs which need not consider the dynamic features.
2.  The second one is that the dynamic features of the USV system can be described by the motion limitator.
3.  The third one is that the trajectory tracking of the TPTH-USV is realized easily by regulating some parameters of the motion limitator and PID controllers.
4.  The last one is that the proposed scheme can be simultaneously used in path following and trajectory tracking, which depends on the constant or variable expected speed of the USV, respectively.

This paper is organized as follows. Section 2 describes the proposed three-layered architecture scheme for trajectory tracking and our TPTH USV called 'Jiuhang 490' USV. Section 3 gives out the implement of the proposed scheme in the three layers. The results of the sea experiments are shown in Section 4 and the conclusions are given in Section 5.

## 2. Three-Layered Architecture Scheme for Trajectory Tracking and 'Jiuhang 490' USV

*2.1. Three-Layered Architecture Scheme*

According to the above three-layer architecture, the trajectory tracking for USVs could be explicitly divided into the trajectory guidance, motion limitator and controller. In the trajectory guidance, an improved LOS was proposed based on an adaptive look-ahead distance which would give the system the desired course and the speed of the USV. In the model limitator, the coupled speed and yaw motion limitator of the USV was acquired based an identification model of the 'Jiuhang490' USV. In the controller, the twin-PID

controller was designed for the course and speed control. The proposed practical trajectory tracking's flow diagram under the three-layered architecture is shown in Figure 2.

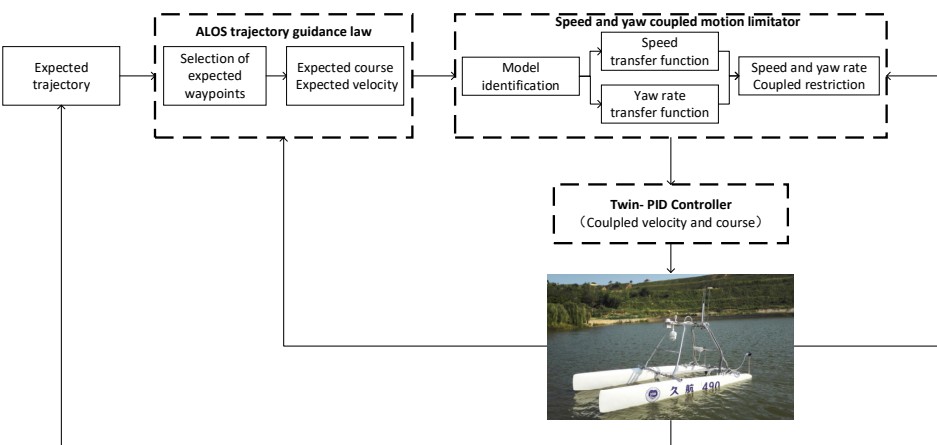

**Figure 2.** The proposed practical trajectory tracking of the USV under three-layered architecture.

## 2.2. 'Jiuhang 490' USV

A TPTH USV called 'Jiuhang 490' was developed in the First Institute of Oceanology, Ministry of Natural Resources in China, in 2017, which is shown in Figure 3. The USV was applied for the offshore emergent observation of nuclear radiation and route monitoring of thermal discharge for national nuclear power stations in our project. The 'Jiuhang 490' USV was 4.9 m in length, 2.5 m in width, 500 kg in weight, the maximal speed was about 5.5 kn, the endurance of the voyage was about 60~80 km and the maximal communication distance was about 10 km. In order to lower the gravity center of the vehicle and to enhance the stability of navigating, the lithium batteries and main control unit was embedded in the fiberglass hulls of the catamaran. Two propellers was used for the stern propulsion, which was controlled by two brushless DC motor actuators separately. Based on an embedded microcomputer, the aboard main control unit was integrated, and a Honeywell HMR3000 digital compass and a Hemisphere VS330 GNSS (Global Navigation Satellite System) compass were adopted for the attitude and position measurements respectively. The planner and controller for trajectory tracking ran in the main control unit. The other integrated sensors contained a gamma detector for nuclear radiation observation, a CTD (Conductivity, Temperature and Depth), a camera and an ultrasonic weather station. More details can be seen in [30].

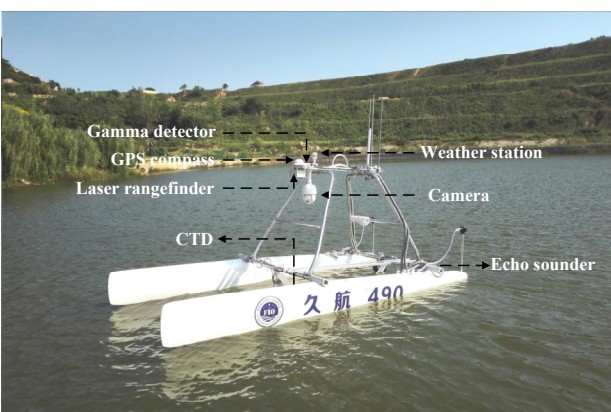

**Figure 3.** The 'Jiu Hang 490' USV and sensors.

## 3. Implement of Trajectory Tracking

### 3.1. Assumptions

To simplify the problem, the motion of the USV in the horizontal plane was considered in this paper. Some assumptions were given out as follows [31]:

- The motion of the USV in roll, pitch and heave directions was neglected, so the motion of the USV was described by three degrees of freedom (DOM), which were surge, sway and yaw.
- The USV had a neutral buoyancy and the origin of the body-fixed coordinate was located at the center of mass.
- The USV was port-starboard symmetric.
- The dynamic equations of the USV did not include the disturbance forces (waves, wind and ocean currents).
- The expected trajectory was of twice continuous differentiability.

### 3.2. Trajectory Guidance Law for Curved Line

In the kinematics layer, the trajectory guidance law for the USV was designed to steer the course and to regulate the speed, which could force the USV to follow the expected trajectory with the temporal constraint, i.e., the tracking position errors $x_{error}$ and $y_{error}$ had to tend to zero in desired moments. In the section, an adaptive look-ahead-based LOS (ALOS) guidance law was designed for the guidance law of curved lines, which would give the expected speed and course for the kinetic layer.

The guidance law for USVs is usually expressed in the body-fixed reference frame o-$X_b Y_b$ {*BF*} and the north-east reference frame O-NE {*NE*}. The look-ahead-based LOS guidance algorithm has usually been used for straight-line path following. In the path following for curved paths without a temporal constraint, there are two solutions, i.e., the first one is that the curved line is divided into some straight-lines, and the other is to minimize the cross-track error in the Serret–Frenet reference frame. The origin of the Serret–Frenet reference frame was set at the position for the shortest distance between the vehicle and the expected curved path. However, the situation was different for the trajectory tracking; for example, the expected waypoint at the certain moment was not coincident with the shortest point between the curved line and the vehicle. Therefore, a reference frame called the Expected Trajectory reference frame {*ET*} was proposed with the origin at the expected waypoint $(x_k, y_k)$ at the k-th moment, where its $Y_{et}$ axis was along the tangential direction for the expected trajectory, and the $X_{et}$ axis was the normal direction. Therefore, it was convenient to calculate the tracking errors $x_e$ and $y_e$ between the USV and the expected trajectory in the reference frame {*ET*}. The reference frame {*ET*} was different from the Serret–Frenet reference frame, where the origin of {*ET*} fixed at the expected waypoint with a temporal constraint and the origin of the Serret–Frenet reference frame changed with the trajectory. The three reference frames and the relationship diagram of trajectory tracking are shown in Figure 4. The points $(x_{k-1}, y_{k-1})$, $(x_k, y_k)$ and $(x_{k+1}, y_{k+1})$ are the three successive expected waypoints of the trajectory at the moment of k − 1, k and k + 1. The point $(x_{los}, y_{los})$ is a virtual expected point calculated by the ALOS algorithm.

#### 3.2.1. Selection of Expected Waypoints

Since a curved trajectory tracking is not different from a straight line tracking, how to select the expected waypoints on the trajectory is an essential step. In order to decrease the calculate cost, there is no need for guidance in every moment in the actual engineering, and the selection of expected waypoints $(x_k, y_k)$ depends on the precision demand of trajectory tracking. The expected curved trajectory is defined as follows,

$$\begin{aligned} x &= x(t), \\ y &= y(t) \end{aligned} \tag{1}$$

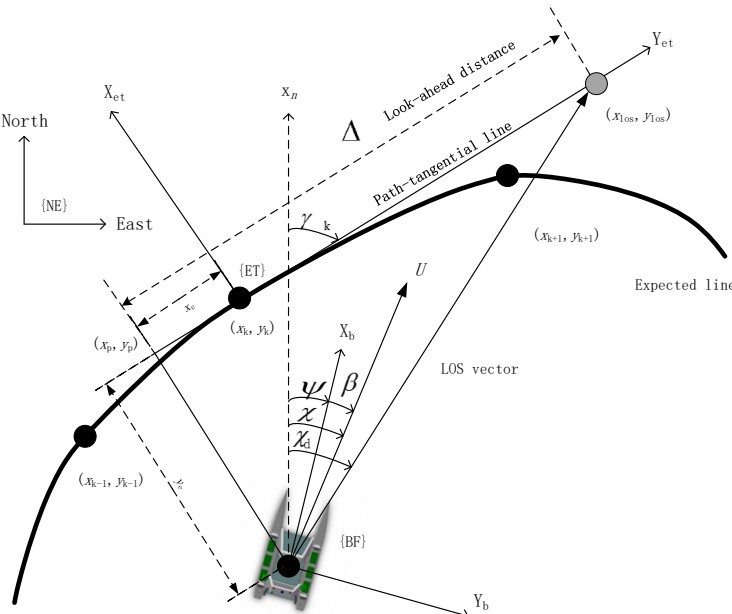

**Figure 4.** The reference frames of USV's trajectory tracking.

The first point of the expected trajectory was set as the first expected waypoint $(x_0, y_0) = (x(1), y(1))$, and the subsequent expected waypoints were selected as follows,

$$(x_k, y_k) = (x(t), y(t)), \text{ if } R(t) < R_0 \text{ or } \dot{U}(t) > U_0, \tag{2}$$

where $R(t)$ is the radius of the curvature for the curved trajectory, $R(t) = \dfrac{\left(\dot{x}^2 + \dot{y}^2\right)^{3/2}}{\dot{x}\cdot\ddot{y} - \ddot{x}\cdot\dot{y}}$ and $U(t) = \sqrt{u(t)^2 + v(t)^2}$, and $u(t) = \dot{x}(t)$, $v(t) = \dot{y}(t)$. The selected rule of the expected waypoints was simple, i.e., when the radius of curvature is smaller than a threshold $R_0$ or the derivative of the expected speed is larger than a threshold $U_0$. The rule assures that the more mutations of the curved trajectory occurring in the space and moment, the more expected waypoints generate.

### 3.2.2. Adaptive LOS Law

According to the transformation relationship between the reference frame {*ET*} and the reference frame {*NE*} in Figure 4, the tracking errors between the USV and the expected curves in {*ET*} are as follows,

$$\begin{bmatrix} x_e \\ y_e \end{bmatrix} = R^T(\gamma_k) \begin{bmatrix} x - x_k \\ y - y_k \end{bmatrix}, \tag{3}$$

where the transformation matrix $R(\gamma_k) = \begin{bmatrix} \cos \gamma_k & -\sin \gamma_k \\ \sin \gamma_k & \cos \gamma_k \end{bmatrix}$, $\gamma_k = \operatorname{atan2}\left(y'_k(\theta), x'_k(\theta)\right) \in [-\pi, \pi]$ is the rotated angle between {*NE*} and {*ET*}, $x_e$ and $y_e$ are the errors of the current position $(x, y)$ and expected waypoint $(x_k, y_k)$ in the reference frame {*ET*}.

The guidance law of the trajectory tracking was to calculate the expected speed $U_d$ and expected course $\chi_d$, which could be designed as according to the conventional LOS guidance algorithm [19],

$$\chi_d = \gamma_k + \arctan\left(-\frac{y_e}{\Delta}\right). \tag{4}$$

In the USV's running or turning in the environment disturbance, such as a wave, there exists a sideslip angle $\beta$ between the heading $\chi$ and the course $\psi$ of the USV in Figure 4. Therefore, the expected heading of the USV for the expected trajectory is as follows,

$$\psi_d = \chi_d - \beta = \gamma_k + \arctan\left(-\frac{y_e}{\Delta}\right) - \beta, \tag{5}$$

where $\Delta$ is the look-ahead distance, and the sideslip angle $\beta = \chi - \psi = \text{atan2}(v, u)$.

In order to improve the tracking performance, the adaptive look-ahead distance $\Delta$ was designed as follows,

$$\Delta = m(1 + 1/|y_e|)L, \tag{6}$$

where m is a gain constant, and L is the length of the USV. It is obvious that when $y_e$ is very small, $\Delta$ is very large. According to the stability proof [18], the larger $\Delta$ is, the more limited the region where the system ULES (Uniform Local Exponential Stability) is becomes. Therefore, $\Delta$ should be restricted when the vehicle is close to the path, and the moderated $\Delta$ was set as n·L, where n is larger than m. If $\Delta > $ n·L, then $\Delta = $ n·L.

In the aspect of the expected speed, it was designed as follows [7],

$$U_d = \frac{(U - P \cdot x_e)\sqrt{y_e^2 + \Delta^2}}{\Delta}, \tag{7}$$

where P is the control gain and U is the cruising speed for the USV. In the restriction level, the speed $U_d$ should be reasonable, so it is moderated, i.e., if $U_d < U_{min}$, $U_d = U_{min}$, and if $U_d > U_{max}$, $U_d = U_{max}$.

The equilibrium points of the cross-track were proven to be globally k exponentially stable [7,18]. It was obvious that the expected speed was a proportional controller in Equation (7), so we adopted a PD controller for the speed term as follows,

$$U_d = \frac{(U - (P \cdot x_e(t+1) + D \cdot (x_e(t+1) - x_e(t))))\sqrt{y_e^2 + \Delta^2}}{\Delta}. \tag{8}$$

### 3.3. Motion Limitator

Since the motion of the USV was considered in the horizontal plane, the speed and yaw rate restrictions were used in the motion limitator corresponding to the two outputs of the trajectory guidance law based on an identified motion model of the USV.

### 3.3.1. Motion Model

The USV's motion model can be described in a plane by three-degrees-of-freedom equations, i.e., the surge, sway and yaw. The transformation relationships between positions and velocities were expressed as follows,

$$\begin{aligned}\dot{x} &= u \cdot \cos(\psi) - v \cdot \sin(\psi) \\ \dot{y} &= u \cdot \sin(\psi) + v \cdot \cos(\psi), \\ \dot{\psi} &= r \end{aligned} \tag{9}$$

where $x$, $y$, and $\psi$ represent the position and orientation in {NE}, and $u$, $v$ and $r$ represent the surge speed, sway speed and yaw rate, respectively, in {BF}.

A general dynamic model was adopted as follows [17],

$$M\dot{v} + C(v)v + D(v)v = \tau, \tag{10}$$

where M represents the inertia matrix, C represents the Coriolis and centripetal matrix, D represents the hydrodynamic drag matrix, and $v$ represents the linear and angular velocity vectors, $\tau$ represents the driven force and the moment of the thrusters. The

above hydrodynamic matrices were given as follows: $M = \begin{bmatrix} m_{11} & 0 & 0 \\ 0 & m_{22} & 0 \\ 0 & 0 & m_{33} \end{bmatrix}, C(v) =$
$\begin{bmatrix} 0 & 0 & -m_{22}v \\ 0 & 0 & m_{11}u \\ m_{22}v & -m_{11}u & 0 \end{bmatrix}$, and $D(v) = \begin{bmatrix} d_{11} & 0 & 0 \\ 0 & d_{22} & 0 \\ 0 & 0 & d_{33} \end{bmatrix} = \begin{bmatrix} X_u & 0 & 0 \\ 0 & Y_v & 0 \\ 0 & 0 & N_r \end{bmatrix}$.

Therefore, the dynamic model of the USV could be described by,

$$\text{Surge}: \ \dot{u} = \frac{m_{22}}{m_{11}}v\cdot r - \frac{d_{11}}{m_{11}}u + \frac{1}{m_{11}}\tau_1, \tag{11}$$

$$\text{Sway}: \ \dot{v} = -\frac{m_{11}}{m_{22}}u\cdot r - \frac{d_{22}}{m_{22}}v, \tag{12}$$

$$\text{Yaw}: \ \ddot{\psi} = \frac{m_{11}-m_{22}}{m_{33}}u\cdot v - \frac{d_{33}}{m_{33}}\dot{\psi} + \frac{1}{m_{33}}\tau_3, \tag{13}$$

where $m_{11}$, $m_{22}$ and $m_{33}$ represent the inertia mass, $d_{11}$, $d_{22}$ and $d_{33}$ represent the drag coefficients, $\tau_1$ and $\tau_2$ represent the thrusts in the $X_b$ and $Y_b$ axes, respectively, and $\tau_3$ represents the thrust moment. It was noted that the value of $\tau_2$ for the TPTH USV equaled to zero, since there was not a propeller or a rudder for the USV in the $Y_b$ axis.

Since the speed $u$ and yaw rate $r$ were the main factors in the model limitator, the model for the surge and yaw motion were identified based on Equations (11) and (13) using the data from a lake trial of the 'Jiuhang 490' USV (Figure 5) on 14–18 September 2017 at a lake in Qingdao city, Shandong Province, China. It is noted that the yaw model was coupled with the speed of the TPTH USV.

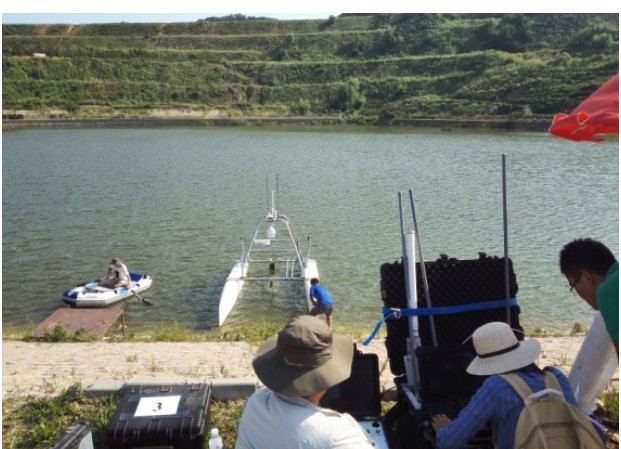

**Figure 5.** The 'JiuHang 490' USV's lake trial in 2017.

### 3.3.2. Model Identification for Surge Motion

The model identification for the surge motion of the USV could be acquired by using the steady state data for a straight-line based on Equation (11),

$$m_{11}\cdot\dot{u} + X_u\cdot u = \tau_1, \tag{14}$$

where $\tau_1 = F_1 + F_2$, $F_1$ and $F_2$ are the thrusts of the left and right propellers, respectively, which ware shown in Figure 6. The relationship between the thrust $\tau_1$ and the basis control variable $C_u$ was fitted linearly by the data from the lake trial which is shown in Figure 7. In Figure 7, the circle represents the measure data, and the solid line represents the linear fitting result. The differential thrust mode was chosen for the TPTH USV as follows,

$$F_1 = k_u\cdot(C_u + C_h), \tag{15}$$

$$F_2 = k_u \cdot (C_u - C_h), \tag{16}$$

where $C_u$ is the basis control variable and $C_h$ is a differential control variable. Therefore, the linear model for the thrusts was as follows,

$$\tau_1 = 2 \cdot k_u \cdot C_u, \tag{17}$$

where $k_u$ is the thrust coefficient for a singular propeller and $k_u = 2.48$ in Figure 7. In the lake trial, the thrust was measured by an ergometer, and the basis control variable $C_u \in [0, 200]$.

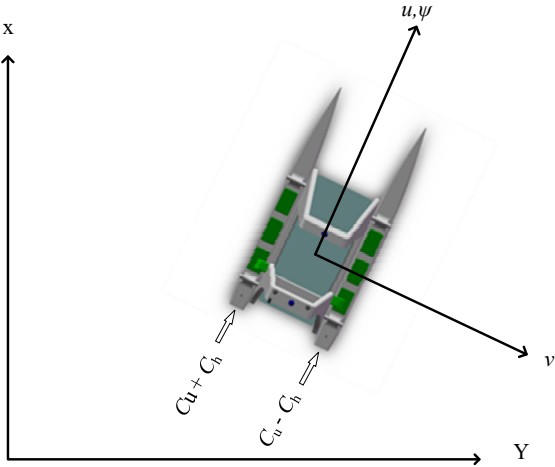

**Figure 6.** The diagram for speed and course regulation of the USV.

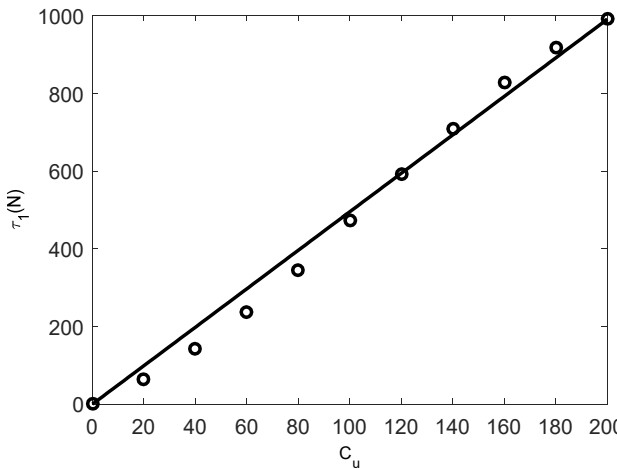

**Figure 7.** The relationship between thrust $\tau_1$ and the basis control variable $C_u$.

The transfer function for the speed $u$ and the basis control variable $C_u$, were acquired by the Laplace transformation based on Equation (14),

$$G_u(s) = \frac{u(s)}{C_u(s)} = \frac{2 \cdot k_u}{m_{11}s + X_u}. \tag{18}$$

Let $K_1 = \frac{2 \cdot k_u}{X_u}$, $T_1 = \frac{m_{11}}{X_u}$, so the transfer function became,

$$G_u(s) = \frac{K_1}{1 + T_1 s} \tag{19}$$

According to the steady state data from the lake trial, the drag coefficient $X_u = \frac{\tau_1}{u} = 359.78$. The inertial mass and added mass were estimated empirically by $m_{11} = m + 0.1 \cdot m = 550$.

Therefore, the transfer function for the speed $u$ and the basis control variable $C_u$ were as follows,

$$G_u(s) = \frac{0.014}{1 + 1.53s} \tag{20}$$

### 3.3.3. Model Identification for Yaw Motion

The yaw model of the USV could be simplified in the condition of the quasistatic course changing based on Equation (13),

$$m_{33}\ddot{\psi} = -d_{33}\dot{\psi} + \tau_3. \tag{21}$$

In the TPTH USV, the steering moment, $\tau_3 = (F_1 - F_2) \cdot d$, and the arm of force d were the perpendicular distance between the propeller and the central line of the USV; then, the steering moment was simply expressed by:

$$\tau_3 = (F_1 - F_2) \cdot d = k_h \cdot C_h. \tag{22}$$

If the nonlinear feature of the propeller was not considered, the thrust of the propeller was simply expressed as:

$$F = k_1 \cdot n^2, \tag{23}$$

where $n$ is the speed of the revolution for the propeller.

The relationship between the speed of revolution and drive voltage for the propeller was as follows,

$$T\frac{dn(t)}{dt} + n(t) = k_2 \cdot v_1(t). \tag{24}$$

Since the propellers for the USV were two small DC motors, the resistance of the armature and moment of inertia were very small, so the temporal parameter T could be neglected; then, the speed of the revolution was as follows,

$$n(t) = k_2 \cdot v_1(t). \tag{25}$$

The relationship between the drive voltage and control voltage of the actuator was simply described,

$$v_1 = k_3 \cdot v_2 \tag{26}$$

and the left and right propellers' control voltages for the actuators were:

$$v_2 = k_4 \cdot (C_u \pm C_h). \tag{27}$$

According to the above relationships, the steering moment was:

$$\tau_3 = k_1 \left( n_1^2 - n_2^2 \right) \cdot d = k_0 \cdot C_u \cdot C_h, \tag{28}$$

where the parameter $k_0 = 4 \cdot k_1 \cdot k_2^2 \cdot k_3^2 \cdot k_4^2 \cdot d$.

Therefore, the steering moment depended not only on a differential variable $C_h$, but also on the basis of the control variable $C_u$, which was more coincidental with the actual situation than that in Equation (22).

In the actual course control, the speed control variable $C_u$ was usually fixed as a constant, so the course control became a single control input issue with a differential control variable $C_h$, which was the same as Equation (22).

Substituting Equation (28) into Equation (21), the relationship between the course $\psi$ and course control variables was,

$$m_{33} \cdot \ddot{\psi} = -d_{33} \cdot \dot{\psi} + k_0 \cdot C_u \cdot C_h. \tag{29}$$

Let $K_2 = \frac{k_0 \cdot C_u}{d_{33}}$ and $T_2 = \frac{m_{33}}{d_{33}}$, and the transfer function for the course control was acquired by the Laplace transformation,

$$G_h(s) = \frac{\psi(s)}{C_h(s)} = \frac{K_2}{s(1 + T_2 s)}. \qquad (30)$$

Remark: The gain $K_2 = \frac{k_0 \cdot C_u}{d_{33}}$ was proportional to motor coefficients $k_0$ and the speed control variable $C_u$, but, inversely, proportional to the rotation drag coefficient $d_{33}$. Therefore, the course control would be affected by the speed control variable. This could be simply understood, because the speed control would affect the course control of the TPTH USV. If the USV ran at a fixed speed, the equation with a fixed parameter could describe the yaw motion. Otherwise, if the USV ran by a variable speed, the course's variance ratio would be proportional to the speed of the USV. The relationship in Equation (30) is similar to the Nomoto model for a conventional ship's steering, and the difference is that the course control is the double-thrust, not a rudder in Nomoto model, and Equation (30) introduces the speed term for the TPTH USV.

It is obvious that Equation (30) is a transformation function with one pole and an integrator. Using the steering data in the lake trial (Figure 8), the transformation function for the course was identified by the System Identification Toolbox (MATLAB) with a gain coefficient $K_2 = 0.14$ and temporal coefficient $T_2 = 0.77$ as follows,

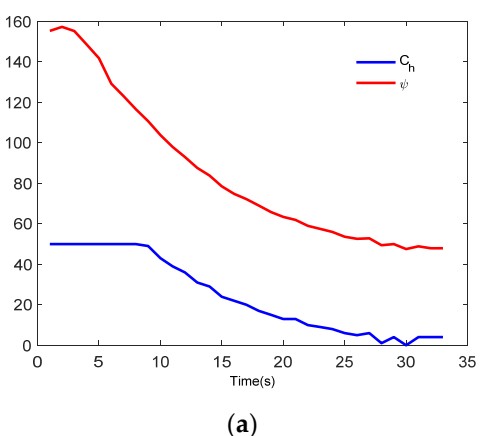

(**a**)

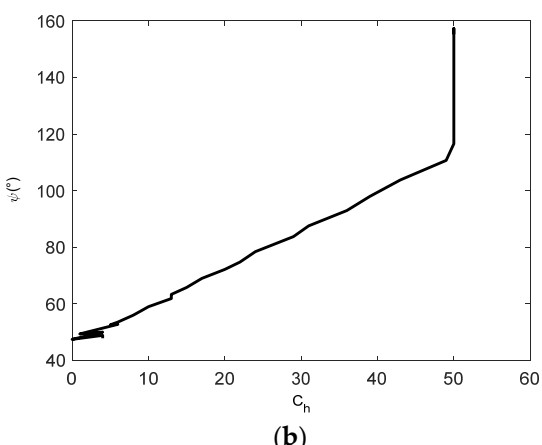

(**b**)

**Figure 8.** Course $\psi$ of the USV with control variable $C_u = 70$ and differential control variable $C_h$ (0~50) (lake data in 2017). (**a**) Course and differential control variable; (**b**) course with control variable.

$$G_h(s) = \frac{0.14}{s(1 + 0.77s)}. \qquad (31)$$

Therefore, the transformation function of the yaw rate could simply be approximated by:

$$G_{yaw}(s) = \frac{0.14}{1 + 0.77s} = C_u \frac{0.002}{1 + 0.77s}. \qquad (32)$$

Equation (32) is a coupling transformation function of the yaw rate with the speed, which was different from the situation for the speed and course control separately. Even though in the situation under a fixed speed, the speed of the USV must change before the speed reaches the fixed value, so the yaw model had to be changed, and it would result in a bad control of course. Therefore, the motion limitator for the yaw rate of the USV can be evaluated by Equation (32), which is related to the speed of the USV. When the expected speed was given, the basic control variable could be evaluated. It was noted that the restriction of the yaw rate varied with the basic control variable for an expected speed. The basis control variable $C_u \in [0, 200]$, so $C_h \in [0, 200 - C_u]$ with a restriction condition

for $C_\mathrm{h} = \min(200 - C_\mathrm{u}, C_\mathrm{u})$. Therefore, we could acquire the restriction of the yaw rate based Equation (32), which is shown in Figure 9.

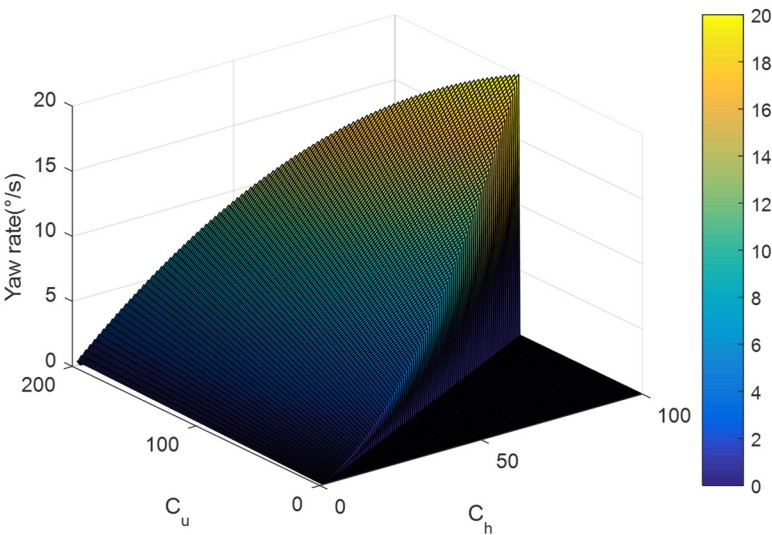

**Figure 9.** The restriction of the yaw rate for the USV.

*3.4. Controllers*

The course and speed regulation of the 'Jiuhang 490' USV was achieved by the two brushless DC motor actuators. The speed of the USV depended on the total thrust from the two propellers, and the course of the USV was adjusted by the thrust difference between the left and right propellers. It is seen that the left and right propellers' control is defined in Figure 6 by:

$$C_\mathrm{l,r} = C_\mathrm{u} \pm C_\mathrm{h}. \tag{33}$$

In Equation (30), at Section 3.3.3, the transformation function of the course was couple with the speed of the USV, and the USV could be seen as a cascade system. Since their relationship is linear, the controllers could be designed by a twin-PID controller, and the diagram for the autonomous control of the USV is shown in Figure 10.

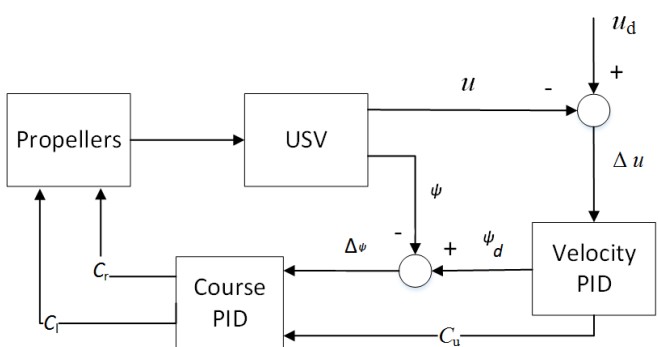

**Figure 10.** The speed and course PID controllers.

The speed controller of the USV was designed as a traditional PID,

$$C_\mathrm{u}(k) = \mathrm{P}_1 \cdot e_\mathrm{u}(k) + \mathrm{I}_1 \cdot \sum_{\mathrm{j}=1}^{k-1} e_\mathrm{u}(j) + \mathrm{D}_1 \cdot (e_\mathrm{u}(k) - e_\mathrm{u}(k-1)), \tag{34}$$

where $e_\mathrm{u}(k)$ is the error between the expected speed and the current speed at the $k$ moment.

The course controller of the USV was designed as an incremental PID,

$$C_\mathrm{h} = \mathrm{P}_2 \cdot (e_\mathrm{h}(k) - e_\mathrm{h}(k-1)) + \mathrm{I}_2 \cdot e_\mathrm{h}(k) + \mathrm{D}_2 \cdot (e_\mathrm{h}(k) - 2e_\mathrm{h}(k-1) + e_\mathrm{h}(k-2)), \tag{35}$$

where $e_h(k)$ is the error between the expected course and the current course at the *k* moment.

Therefore, the uncoupling course PID controller was,

$$C_l(k) = C_u + P_2 \cdot (e_h(k) - e_h(k-1)) + I_2 \cdot e_h(k) + D_2 \cdot (e_h(k) - 2e_h(k-1) + e_h(k-2)),$$
$$C_r(k) = C_u - P_2 \cdot (e_h(k) - e_h(k-1)) + I_2 \cdot e_h(k) + D_2 \cdot (e_h(k) - 2e_h(k-1) + e_h(k-2)). \quad (36)$$

Based on the transformation function of the speed in Equation (20), the parameter tuning was executed by the cut-and-trial method, and the parameters were acquired as $P_1 = 30.0$, $I_1 = 30.0$ and $D_1 = 3.0$. Based on the transformation function of the course control Equation (31), the parameter tuning was executed by the Ziegler–Nichols frequency response method, and the parameters were acquired as $P_2 = 66.7$, $I_2 = 24.1$ and $D_2 = 46.2$. In the low level control of the actuators for the USV's propellers, the control voltage for the actuators was described according to the intrinsic performance of the propeller in the sea trials as follows,

$$V_{l,r} = \begin{cases} 1 - \left(\frac{0.8}{200}\right) \cdot C_{l,r}, & \text{when the propellor is corotation} \\ 1.4 + \left(\frac{0.8}{200}\right) \cdot C_{l,r}, & \text{when the propellor is reverse} \end{cases} , C_{l,r} \in [0, 200], \quad (37)$$

where the stop voltage of the actuators is 1.0 Volt and the control dead zone of the actuators is about 0.2 Volt.

## 4. Result of Sea Experiments

The proposed trajectory tracking scheme was tested in sea experiments using our 'Jiuhang490' USV. The sea experiments were executed at Nanjiang dock in Qingdao City, China, on 16–31 July 2018. During the sea experiments, the hardware system, autonomous control and data acquisition for nuclear radiation were tested [30], and the sea experiments for the USV are shown in Figure 11.

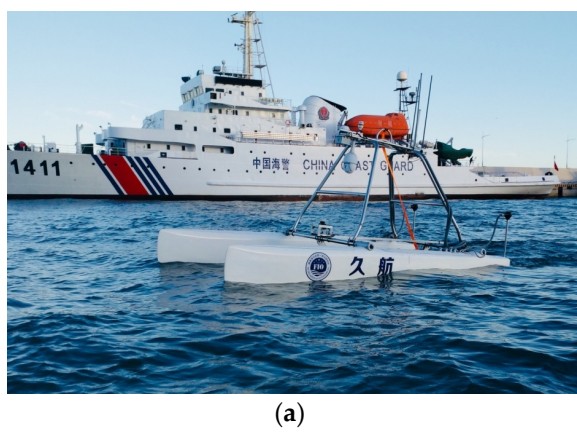
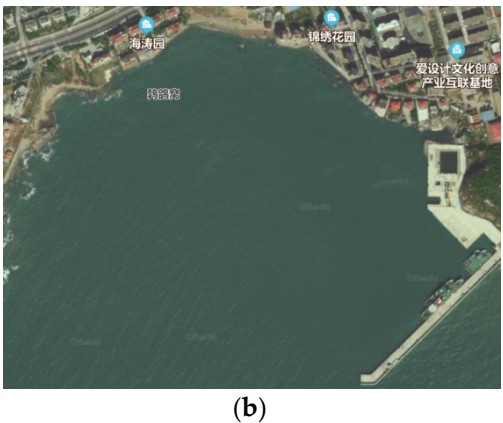

| (a) | (b) |

**Figure 11.** Sea experiments at Qingdao in 2018. (**a**) The 'Jiuhang' USV during the sea experiments; (**b**) location of the sea experiments.

### 4.1. Dynamics Control Results

The autonomous control for the expected course and speed is shown in Figure 12. In Figure 12a, the course and speed of the USV followed well with the expected course of 220° and the expected speed of 4 kn. The initial course was about 269°, and the initial speed was zero. The trial result denoted that the coupled controllers for the course and speed were effective; however, there existed a fluctuation in some tracking errors. There were three reasons for the fluctuation; the first one was that the variable attitude of the USV caused by waves led to a fluctuation in the course's measurement by the digital compass and speed's measurement by the GPS, the second one was that the circumstance compensation for the controllers was not considered, and the third one was that the precision of the speed was about 0.1 kn. Though there were some fluctuations in the following error, the following result for the USV was stable in the corresponding trajectory in Figure 12b, where the circle and the plus denote the initial position and the terminal position of the USV, respectively.

The performance of the coupled controllers for the expected course of 330° and expected speed of 5 kn was good, which can be seen in the following results in Figure 13.

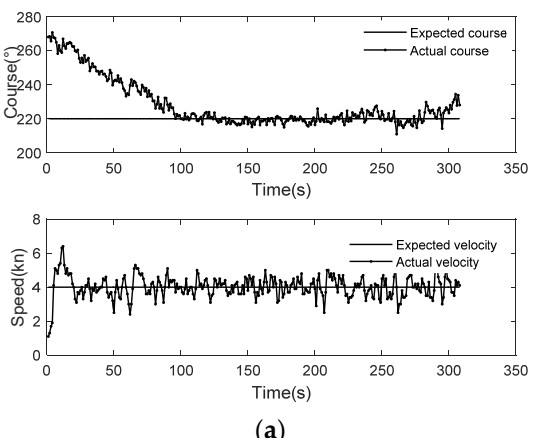 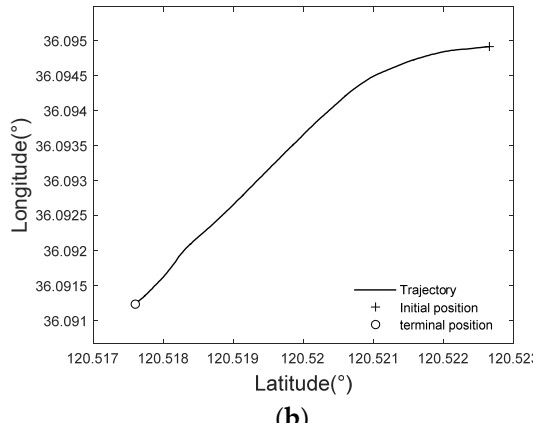

(**a**)  (**b**)

**Figure 12.** The course and speed control of the USV with expected course (220°) and speed (4 kn). (**a**) The coupled control for course and speed; (**b**) the corresponding trajectory of the USV.

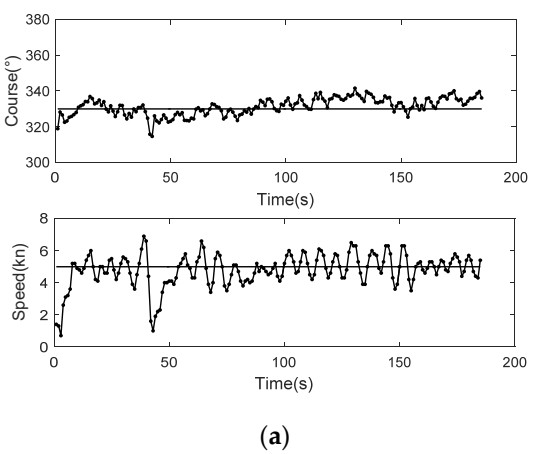 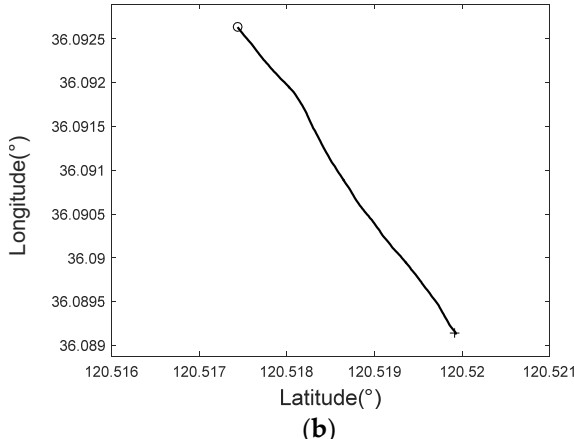

(**a**)  (**b**)

**Figure 13.** The course and speed control of the USV with expected course (330°) and speed (5 kn). (**a**) The coupled control for course and speed; (**b**) the corresponding trajectory of the USV.

In order to test the course and speed coupled controllers, seven autonomous courses of running of the USV were performed in the sea trials. Without the loss of generality, the seven expected courses of the USV were designed in four quadrants, and the corresponding expected velocities were set between 1 kn and 5 kn. The following errors of the course and speed in the stability running of the USV are shown, respectively, in Table 1. In order to reduce the control frequency for the propellers, the control precisions for the course and speed following were set as 0.5° and 0.1 kn, respectively, which equaled to the measurement precision of the course by the HMR 3000 digital compass and to the measurement precision of the speed by Hemisphere VS330 GPS onboard the USV, respectively. When the course and speed of the USV reached the control precision, controls for the course and speed were stopped.

In Table 1, the RMSEs of course tracking were between 3.5° and 7.3° and the RMSEs of speed tracking were between 0.4 kn and 1.1 kn, except for case seven with the lowest expected speed of 1 kn. Though the experiment was carried out in the port, there always existed a disturbance of the ocean environment, such as wind, current and wave, so the RMSEs of the course and speed tracking were accepted. The tracking performance of case seven for the course was very bad, because the USV was very difficult to steer at a low speed.

**Table 1.** RMSE of course and speed coupled control during the sea experiments.

| No. | Expected Course/Speed | RMSE of Course Control | RMSE of Speed Control |
|:---:|:---:|:---:|:---:|
| 1 | 330°/5 kn * | 5.0° | 1.1 kn |
| 2 | 220°/4 kn * | 3.5° | 0.5 kn |
| 3 | 315°/4 kn | 5.2° | 0.7 kn |
| 4 | 280°/3 kn | 5.7° | 0.4 kn |
| 5 | 130°/2 kn | 5.3° | 0.5 kn |
| 6 | 10°/2 kn | 7.3° | 0.7 kn |
| 7 | 330°/1 kn | 30.7° | 0.3 kn |

* Cases 1 and 2 were results of the course and speed control in Figures 12 and 13.

### 4.2. Trajectory Tracking Results

In order to test the trajectory tracking scheme of the USV, the line and rectangle trajectories tracking were achieved by the 'Jiuhang' USV in the sea experiments, and the typical results are shown in Figures 14 and 15.

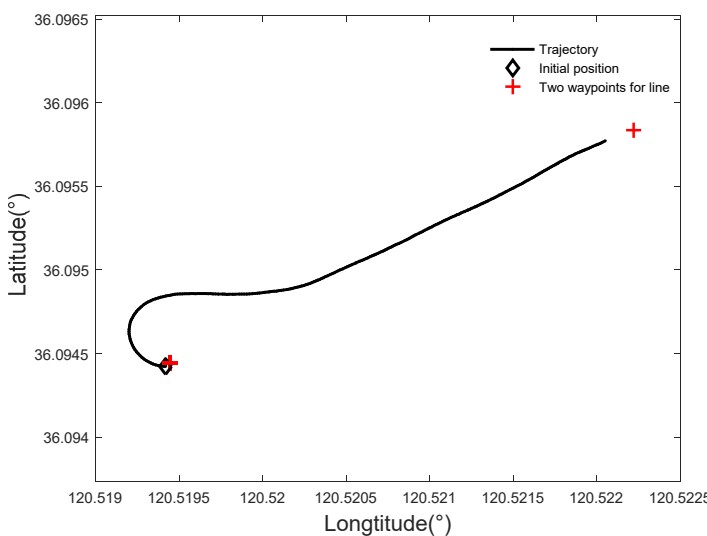

**Figure 14.** Line tracking by the USV in the sea experiment.

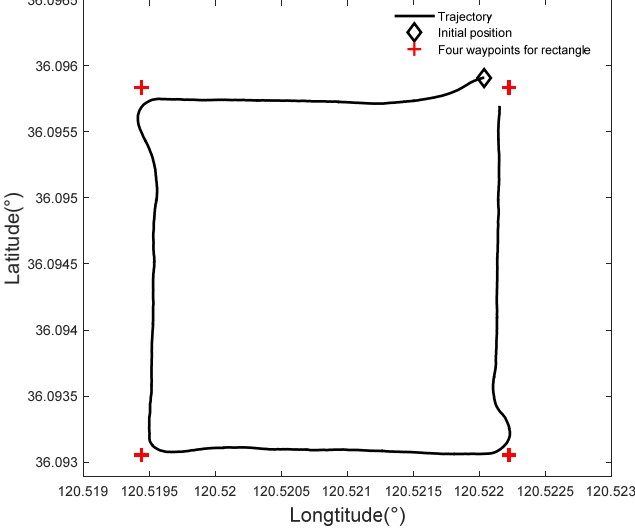

**Figure 15.** Rectangular trajectory tracking of the USV in the sea experiment.

In Figure 14, trajectory tracking for a line was performed, where the red plusses represent the two waypoints of the expected line, and the black diamond represents the

initial position of the USV, and the black line is the trajectory of the USV. It was shown that the initial course of the USV was about 266.1°, which was almost opposite to the expected direction in Figure 14, so the USV could track the line well by a large angle turning. It seemed that the USV did not reach the end point, thanks to an arriving radius around the end point being set. The speed was about 3.3 kn when trajectory tracking of the line was stable. The voyage distance of the USV was about 290.0 m, and the length of the expected line was about 249.6 m.

A rectangular trajectory for four waypoints was tracked by the USV, which is shown in Figure 15, where the red plusses represent the four waypoints of the expected rectangle, the black diamond represents the initial position of the USV, and the black line is the trajectory of the USV. The achieved range between the USV and the current expected waypoint is set as 5 m, i.e., when the USV reach to the range, the tracking for the current waypoint was finished and the USV turned to the next waypoint. It was shown that the performance of trajectory tracking was good except for some draft in the vertexes of the rectangle due to no special disposing for plan trajectory around the vertexes. In Figure 15, the rectangle was about 249.6 m × 308.8 m, and the voyage distance of the USV was about 1200.0 m. There was some offset between the expected trajectory and the actual trajectory, and the one reason was that the precision of the GPS was about 2.5 m and the orientation precision of the digital compass was about 0.5°; the other reason was that the USV's control was disturbed by the wind and waves.

## 5. Conclusions

In view of practical engineering, a three-layered architecture for TPTH-USV's trajectory tracking was proposed and validated using the 'Jiuhang' USV in the sea experiments. Besides the conventional kinematic and kinetic layer, a motion restriction layer was added in the three-layered architecture. The proposed guidance law and controllers in the first and third layers were properly suitable for the type of TPTH USVs, which could be applied directly without considering the motion model's variety. The ALOS law can force the USV to track a curved line with a time constraint and give out speed and course variables which are taken as the expected value in the third layer. The twin PID controller can justly solve the speed and course coupled issue of the TPTH USV. The identified model of the USV was used to restrict the basis control variable and differential control variable simply in the motion limitator. In the future, the three-layered architecture of the TPTH-USV will be improved considering sea disturbances, such as waves and current.

**Author Contributions:** Conceptualization, J.J.; methodology, J.J.; software, D.L.; validation, D.W.; formal analysis, D.L.; investigation, D.L.; resources, Y.M.; data curation, D.L.; writing—original draft preparation, J.J.; writing—review and editing, J.J.; visualization, D.W.; supervision, Y.M.; project administration, Y.M.; funding acquisition, Y.M. All authors have read and agreed to the published version of the manuscript.

**Funding:** This research was funded by the National Key Research and Development Program of China, grant number 2017YFC14052.

**Institutional Review Board Statement:** Not applicable.

**Informed Consent Statement:** Not applicable.

**Data Availability Statement:** Not applicable.

**Acknowledgments:** The authors would like to thank Feng Shao and Junnan Shi at FIO for experimental help.

**Conflicts of Interest:** The authors declare no conflict of interest.

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
