# Peer review of "A Practical Trajectory Tracking Scheme for a Twin-Propeller Twin-Hull Unmanned Surface Vehicle"

_jmse, doi:10.3390/jmse9101070_

Round 1

Reviewer 1 Report

I am confident that the topic of the article is interesting and essential in terms of research however, some content presented is not clear enough. In order to enhance the article quality, I would suggest major revision as follow:

1/ Despite the motivating topic and the paper seems to have a contribution. However, the novelty of the approach is rather low since it seems all the methods are existing in the guidance part and the PID control is simple and basic algorithm.

2/ The quality of the introduction can be improved. For example, more references of point stabilization, path following, and trajectory tracking of marine vehicles need to be added and the concept of sliding mode control, PID, and backstepping should be discussed in terms of a deeper state of the art, some new works related to sliding mode control of marine vehicles, especially the dynamic sliding mode control methods, robust sliding mode method, multiple sliding mode methods, and so on, should be included. To help the authors in this direction, I suggest the following reference: https://doi.org/10.3390/s21030747, DOI: 10.1109/ACCESS.2020.3048706, https://doi.org/10.3390/s20092633, https://doi.org/10.3390/s20051329, https://ojs.imeti.org/index.php/PETI/article/view/2892,  https://doi.org/10.5574/JAROE.2016.2.4.192.

And the introduction should be added to do a better job of explaining the existing methods and why they are or are not valuable.

3/ Thereare some overlap in text in the paper (same phrases and sentences). It shouldn't be cut-and-paste. For example, lines 44-45 and lines 78-79 have same sentences, “Many control laws are designed…. Out perfectly”, lines 101-103, and lines 131-133, “an improved LOS is ……. of the USV”. The authors should check all paper again.

4/ The abbreviation GPS in section 2.2 must be explained.

5/ In section 3.1.2 Adaptive LOS law, the authors proposed an improved LOS algorithm. However, what are the differences or improved points of the LOS algorithm in this paper compared to LOS algorithm in references [6] and [17]? Please clarify it!

6/ In section 3.2, a sub-section “Assumptions” should be added to make the problem clearer. All assumptions and physical constraints should be provided. How many degrees of freedom DOF is applied in the USV system? The author can refer to the “Assumptions” section of the following paper: https://doi.org/10.3390/s20051329

7/ How to define the thrust coefficient k_u=2.48 as the authors did? Also how to define all coefficients k1, k2, k3, and k4 in Equations 25-28? Please explain in more detail.

8/ In section 3.3, how to get Equation 37? Please clarify this.

9/The parameters for simulation such as hydrodynamic coefficients of the system and parameters for disturbances (wind, wave, and current force) need to be added in the paper.

10/ In section 4.1, the authors mentioned that “considering the disturbances of ocean….” (lines 464-465). Did the authors consider the disturbance acting on the system? What happens if the effects of water current disturbance are included in the proposed method? The reviewer did not see disturbance results in the paper. Furthermore, please check the sentence “the tracking performance…. low speed (lines 466-467)”, its meaning is not clear and confused “different” or “difficult”.

11/ Simulation analysis is insufficient, more simulation results need to be added in the paper, such as the propeller forces. Especially, in section 4.2, some simulation results of positions of USV (x, y, yaw angle with respect to time) as well as the thruster forces should be mentioned in the simulation result part. The explanations and analysis of simulation results should be enriched to show the validity of the data.

12/ The paper would benefit greatly if it were expanded with a comparison of the proposed method with other algorithms known in the literature. Results are not compared with other approaches and it is difficult to assess whether the proposed approach is better than the existing ones.

13/ Conclusion needs extended elaboration on the topic, results, lessons learned, and future works.

14/ The English writing of this paper should be thoroughly polished, especially some grammatical errors and formula format errors are required to be revised carefully. For example, in line 88, “USVS” should be “USVs”, in line 340, “the Eq. (22)” should be “Eq. (22)”, in line 343, “the Eq. (22)” should be “Eq. (22)” in line 359, “the Eq. (30)” should be “Eq. (30)”, in line 371, “The Eq. (32)” should be “Eq. (32)”, in line 390, “the Eq. (30)” should be “Eq. (30)”, in line 427, “the Figure 12(a)” should be “Figure 12(a)”, in line 429, “denote” should be “denotes”, in line 435, “is” should be “are”, in line 436, “ the Figure 12(b)” should be “Figure 12(b)”, and so on.

Author Response

Dear Editor,

  First of all, we sincerely thanks for your significance comments. According to your comments, the explanations and revision are given out as follows.

Question 1. Despite the motivating topic and the paper seems to have a contribution. However, the novelty of the approach is rather low since it seems all the methods are existing in the guidance part and the PID control is simple and basic algorithm.

Response 1. Thanks for your comment. Although the approach of the manuscript look likes very usual, the proposed three-layered architecture of trajectory tracking is designed for twin-propeller and twin-hull USV in view of practical application, which has been validated in the sea experiment. The authors don’t try to raise a novel control law, but try to give a universal solution scheme for trajectory tracking of the special type of USV.

Question 2. The quality of the introduction can be improved. For example, more references of point stabilization, path following, and trajectory tracking of marine vehicles need to be added and the concept of sliding mode control, PID, and backstepping should be discussed in terms of a deeper state of the art, some new works related to sliding mode control of marine vehicles, especially the dynamic sliding mode control methods, robust sliding mode method, multiple sliding mode methods, and so on, should be included. To help the authors in this direction, I suggest the following reference: https://doi.org/10.3390/s21030747, DOI: 10.1109/ACCESS.2020.3048706, https://doi.org/10.3390/s20092633, https://doi.org/10.3390/s20051329, https://ojs.imeti.org/index.php/PETI/article/view/2892,  https://doi.org/10.5574/JAROE.2016.2.4.192.

And the introduction should be added to do a better job of explaining the existing methods and why they are or are not valuable.

Response 2. Thanks for your comment. As said in the manuscript that many control laws are designed based on backstepping technology, and the stabilization are usually given out perfectly. So we intend to introduce the state of art for the tracking scheme in practical view, but don't intend to elaborate the whole control laws especially by complicated theories. By contrast, the sliding mode method is a kind of practical algorithm, and we add and quote the suggested reference in the manuscript.

Question 3. There are some overlap in text in the paper (same phrases and sentences). It shouldn't be cut-and-paste. For example, lines 44-45 and lines 78-79 have same sentences, “Many control laws are designed…. Out perfectly”, lines 101-103, and lines 131-133, “an improved LOS is ……. of the USV”. The authors should check all paper again.

Response 3. Thanks for your comment. There are some overlap in the manuscript indeed, which are deleted in the revised manuscript.

Question 4. The abbreviation GPS in section 2.2 must be explained.

Response 4. The abbreviation GPS in section 2.2 is explained in the revised manuscript.

Question 5. In section 3.1.2 Adaptive LOS law, the authors proposed an improved LOS algorithm. However, what are the differences or improved points of the LOS algorithm in this paper compared to LOS algorithm in references [6] and [17]? Please clarify it!

Response 5. Thanks for your comment. There are three improvement points of the LOS algorithm in the manuscript compared to LOS algorithm in the two references.

1) The expected waypoints for trajectory tracking are selected adaptively based on the radius of curvature for the curved trajectory and expected speed using Eq(2), not set in advance. The selected rule of the expected waypoints is simple, i.e. when the radius of curvature is smaller than a threshold R0 or the derivative of the expected speed is larger than a threshold U0. The rule can assure that the more mutation of the curved trajectory occurs in space and moment, the more expected waypoint generate.

2) A adaptive look-ahead distance  is designed which is different from the two reference, i.e. , and the expression is simple and valid.

3) The expected speed is a proportional controller in references [6] and [17], so we adopt a PD controller for speed term as follows, .         

Question 6. In section 3.2, a sub-section “Assumptions” should be added to make the problem clearer. All assumptions and physical constraints should be provided. How many degrees of freedom DOF is applied in the USV system? The author can refer to the “Assumptions” section of the following paper: https://doi.org/10.3390/s20051329

Response 6. Thanks for your comment. Assumptions from the paper are added in the advised manuscript, which are as follows,

  • The motion of the USV in roll, pitch, and heave directions is neglected, so the motion of the USV is described by 3 degrees of freedom (DOM), which is surge, sway and yaw.
  • The USV had neutral buoyancy and the origin of the body-fixed coordinate is located at the center of mass.
  • The USV is port-starboard synmetric.
  • The dynamic equations of the USV don't include the disturbance forces (waves, wind, and ocean currents).
  • The expected trajectory is twice continuous differentiability.

Question 7. How to define the thrust coefficient k_u=2.48 as the authors did? Also how to define all coefficients k1, k2, k3, and k4 in Equations 25-28? Please explain in more detail.

Response 7. Thanks for your comment. A linear relationship between the thrust of the propeller and the control variable is found in the lake data, so we define the thrust coefficient as the linear regression coefficient. As the same in Equations25-28, we assume the relationships are linear.

Question 8. In section 3.3, how to get Equation 37? Please clarify this.

Response 8. Thanks for your comment. The Equation 37 denotes the relationship between the control voltage and the control variable in the bottom control. For example, when the propellor is corotation, the larger the control variable  is, the smaller the control voltage is, and the larger rotated speed of the propellor is. In contrast, when the propellor is reverse, the larger the control variable  is, the larger the control voltage is, and the larger rotated speed of the propellor is. The Equation 37 is defined by the propellor’s identity and our control design.

Question 9. The parameters for simulation such as hydrodynamic coefficients of the system and parameters for disturbances (wind, wave, and current force) need to be added in the paper.

Response 9. Thanks for your comment. The hydrodynamic coefficients are identified by the lake data. The feasibility and practicability of the proposed trajectory tracking scheme is validated through sea experiments at Nanjiang dock in Qingdao City, China on 16-31 July, 2018, and simulation isn't executed.

Question 10. In section 4.1, the authors mentioned that “considering the disturbances of ocean….” (lines 464-465). Did the authors consider the disturbance acting on the system? What happens if the effects of water current disturbance are included in the proposed method? The reviewer did not see disturbance results in the paper. Furthermore, please check the sentence “the tracking performance…. low speed (lines 466-467)”, its meaning is not clear and confused “different” or “difficult”.

Response 10. The disturbance isn't considered in tracking scheme, and the expression isn't exact in the manuscript and is corrected in the revised manuscript. In the sea experiment, the control results is well under the sea disturbance though we don't measure the sea disturbance, such as current, wave. As you said, the "different" should be difficult.

Question 11. Simulation analysis is insufficient, more simulation results need to be added in the paper, such as the propeller forces. Especially, in section 4.2, some simulation results of positions of USV (x, y, yaw angle with respect to time) as well as the thruster forces should be mentioned in the simulation result part. The explanations and analysis of simulation results should be enriched to show the validity of the data.

Response 11. Thanks for your comment. All the data are derived from the sea experiments, not simulations. The thruster forces haven't been measured in the sea experiments. The explanations and analysis of experiment results are enriched in the revised manuscript.

Question 12. The paper would benefit greatly if it were expanded with a comparison of the proposed method with other algorithms known in the literature. Results are not compared with other approaches and it is difficult to assess whether the proposed approach is better than the existing ones.

Response 12. Thanks for your comment. The proposed scheme has been tested in sea experiments, however unfortunately there is no comparison with the other algorithms, and this is our future job.

Question 13. Conclusion needs extended elaboration on the topic, results, lessons learned, and future works.

Response 13. The conclusions are enriched in the revised manuscript according to your advices.

Question 14. The English writing of this paper should be thoroughly polished, especially some grammatical errors and formula format errors are required to be revised carefully. For example, in line 88, “USVS” should be “USVs”, in line 340, “the Eq. (22)” should be “Eq. (22)”, in line 343, “the Eq. (22)” should be “Eq. (22)” in line 359, “the Eq. (30)” should be “Eq. (30)”, in line 371, “The Eq. (32)” should be “Eq. (32)”, in line 390, “the Eq. (30)” should be “Eq. (30)”, in line 427, “the Figure 12(a)” should be “Figure 12(a)”, in line 429, “denote” should be “denotes”, in line 435, “is” should be “are”, in line 436, “ the Figure 12(b)” should be “Figure 12(b)”, and so on.

Response 14. Thanks for your comment. All the English writing of the paper are corrected and polished in the revised manuscript.

All the revision are denoted using the“Track Changes” function in the manuscript.

Reviewer 2 Report

The article concern a very important problem in this domain. The approach presented can solve the problem for specific type of vessel. This needs to be clarify in the paper. Also, there are some recent relevant research in this field that worth mentioning in the related work such as "Using deep reinforcement learning methods for autonomous vessels in 2d environments". The language in the paper is reasonably good but it needs to be proof read before final submission.

Author Response

Dear Reviewer,

  First of all, we sincerely thanks for your significance comments. According to your comments, the explanations and revision are given out as follows.

Question. The article concern a very important problem in this domain. The approach presented can solve the problem for specific type of vessel. This needs to be clarify in the paper. Also, there are some recent relevant research in this field that worth mentioning in the related work such as "Using deep reinforcement learning methods for autonomous vessels in 2d environments". The language in the paper is reasonably good but it needs to be proof read before final submission.

Response. Thanks for your comment. The reference is very suitable for the paper, and it is quoted. The English writing is polished in the revise manuscript. Other revision are denoted using the“Track Changes” function in the manuscript, which include introduction, assumption, conclusion and so on.

Round 2

Reviewer 1 Report

In a general way, most of my comments were answered by the authors. The manuscript now is acceptable for publishing.